Mapping the distribution of scale-rayed wrasse Acantholabrus palloni in Swedish Skagerrak using angling records

http://orcid.org/0000-0003-1091-2225 Näslund Joacim 1 joacim.naslund@gmail.com
Lundgren Markus 2
1 Department of Zoology, Stockholm University , Stockholm , Sweden
2 Swedish Anglers Association , Gothenburg , Sweden
Arlinghaus Robert
Electronic publication date: 2018 Nov 6
Publication date: 2018
Volume: 6
Electronic Location ID: e5900
Received 2018 Mar 2; Accepted 2018 Oct 2
Copyright: © 2018 Näslund and Lundgren
Copyright year: 2018
Copyright holder: Näslund and Lundgren
License: This is an open access article distributed under the terms of the Creative Commons Attribution License, which permits unrestricted use, distribution, reproduction and adaptation in any medium and for any purpose provided that it is properly attributed. For attribution, the original author(s), title, publication source (PeerJ) and either DOI or URL of the article must be cited.
License URL: https://creativecommons.org/licenses/by/4.0/

Keywords: Distribution, Acantholabrus palloni, Labridae, Angling records, Skagerrak, Citizen-generated data

Funding: The authors received no funding for this work.

==============================
In this paper, we map the distribution of scale-rayed wrasse Acantholabrus palloni in eastern Skagerrak based on a combination of verified and personally communicated angling records. Long thought to be occasional vagrants outside its known range in the eastern Atlantic Ocean and Mediterranean Sea, we ask if this rare and understudied labrid has expanded its range and become established in Swedish waters. A recent surge in verified angling records in the Swedish Anglers Association’s specimen database Storfiskregistret provides information to suggest that this species should no longer be considered an occasional guest, but rather a species established in the Swedish parts of Skagerrak. These records are supported by additional personal communications with anglers. The species is currently well spread geographically along the Swedish Skagerrak coast, with many locations providing repeated captures of adult fish over multiple years. The typical Swedish catch sites are rocky reefs located between the general 40- and 80-m depth curves, likely influenced by currents bringing higher-salinity water from the North Sea. The present study show that angling records can provide an important, but underutilized, resource for mapping the distribution of data-deficient fish species.

Background

Records of rare species and their natural history can constitute important information for future research on these species and the ecosystems they occur in, for example, with respect to human impacts (Boero, 2013; Able, 2016). Obtaining records of rare species is, however, time consuming and professional biologists are often active within projects limited in time and space, making alternative sources for information important (Devictor, Whittaker & Beltrame, 2010; Bradter et al., in press). This is particularly true for records from aquatic environments, where the occurring species rarely can be observed directly, but rather have to be obtained by indiscriminate techniques such as trawls, nets, and dredges, which are also limited in their areal coverage at any given point in time. To achieve a larger spatial and temporal coverage, there is typically need for larger efforts and manpower than normally possible within a normal research project. Non-professional experts can help in the collection of species occurrence data, a feature which has been widely utilized for a long time within the scientific field of ornithology (Silvertown, 2009; Dickinson, Zuckerberg & Bonter, 2010; Bradter et al., in press). However, non-biologists well versed in species determination, -ecology and -distributions, exist not only for birds, but also for fish in the form of leisure anglers (Granek et al., 2008). Angling is a common leisure activity in large parts of the world (Arlinghaus, Tillner & Bork, 2015; Hyder et al., 2018) and anglers often document (photographs and personal journals; e.g., Banha, Ilhéu & Anastácio, 2015; Skov, Jansen & Arlinghaus, 2017) and report their catches (public or closed databases; e.g., Venturelli, Hyder & Skov, 2017; this paper: Materials and Methods). Some anglers are also specializing in “collecting” different species (a similar concept to “twitching” in the bird-watching community; see, e.g., the Swedish online angling community “50-klubben”; “The club of 50 species,” http://www.50klubben.se/). Information about rare species’ occurrence is also commonly spread openly within the angler community (Lundgren & Waje, 2015). Hence, anglers’ records and notes are excellent sources for confirmed qualitative data on presence of species in certain areas and can be useful and important, but underutilized, auxiliary, and corroborating sources for mapping distributions and habitats of data-deficient species (Fetterplace et al., 2018). In this paper, the aim is to present a desktop study where citizen-generated data, in the form of private and publicly available angling records, are used to map out a tentative distribution map for a data-deficient species, the scale-rayed wrasse Acantholabrus palloni (Risso, 1810), in Swedish waters.

Acantholabrus palloni is a labrid fish inhabiting the eastern Atlantic Ocean and the Mediterranean Sea, with a known latitudinal range from Gabon in western Africa to mid- Norway, including areas around some of the eastern Atlantic offshore islands, like Madeira and the Canary Islands (Andersson, 1942; Debelius, 1997; Muus, Nielsen & Svedberg, 1999; Pollard, 2010; Kullander et al., 2012). The full range of the species is continuously being mapped out, with relatively recent documented records from, for example, the Azores and Cape Verde Islands (Santos, Porteiro & Barreiros, 1997; Wirtz et al., 2013). The species is considered rare throughout its known range, but since its typical habitat (coralline and rocky offshore reefs) are seldom trawled, its population may be underestimated due to lack of capture records (Swaby & Potts, 1990; Pollard, 2010). Little is known about its ecology; it is considered to live solitarily or in small groups, and the diet mainly consists of benthic invertebrates (Pollard, 2010). It is light brown in color and characterized by one black blotch on the posterior part of the dorsal fin, a black blotch on the dorsal part of the trunk and several lighter blotches on the back, below the dorsal fin. By these characteristics, the species is well distinguished from other wrasse species in Swedish waters. In the Mediterranean Sea it is often found on rocky bottoms at depths below 80 m (Sartoretto et al., 1997). In the northern parts of its range, however, it has been noted at shallower depth (from 18 m; Debelius, 1997; Kullander et al., 2012).

In Norway, the species has been considered rare, albeit potentially yearly in occurrence (Curry-Lindahl, 1985), but recent evidence suggest that there are larger concentrations of the species in, for example, the Hardangerfjord, and anecdotal reports from scuba-divers suggest it is more common than previously thought (Espeland et al., 2010). The species is regularly captured by anglers in the Norwegian part of northern Skagerrak, just south of Langesund, on rocky bottoms at 40–60 m depth, elevating from deeper soft bottoms (Fig. 1., position 1; M. Lundgren, 2017, personal observations; also documented in the catch-records of the Langesund Seafood and Fishing Festival; http://www.lsff.no/; data accessed 23 April 2014). The first recorded Scandinavian specimen (recorded as A. Couchi) was retrieved in 1869 from around 50 m depth in the area around the island Hidra (Hitterø) close to Flekkefjord in south-western Norway, and is preserved at the Swedish Museum of Natural History (cat. no. NRM 47556) (Öberg, 1870; SMNH (Swedish Museum of Natural History), 2018; description reiterated in Stuxberg, 1894). This specimen was confirmed as an adult individual belonging to the species Labrus palloni (junior synonym to A. palloni) by Lilljeborg (1881). Another three preserved specimens from the Norwegian parts of the eastern Skagerrak (captured in 1966, 1968, and 1985), around the southern parts of the Oslofjord area (Fig. 1., one specimen at position 2, and two specimens at position 3), are available at the Natural History Museum, University of Oslo, according to the Global Biodiversity Information Facility (GBIF) database (https://www.gbif.org/). In addition, Andersson (1942) notes that a few specimens had been caught between Stavanger and Kristiansand in Norway.

Figure 1 Positions for records of Acantholabrus palloni in Skagerrak.

Red dots represent non-angling records, showing the previously documented occurrence, and blue dots represent angling records reported in this paper. Red arrows show the large-scale current patterns in the area, and dotted lines delineate approximate 40- and 80-m depth curves. A: Hidra Island, Norway, first Scandinavian record (Öberg, 1870); B: Singlefjord mouth, Sweden, first Swedish record (identical to position 5) (Cedhagen & Hansson, 1995); 1: S. Langesund city, Norway; 2: Ferder Lighthouse, Oslofjord, Norway; 3: S. Missingen Islands, Oslofjord, Norway; 4: Koster Fjord area, Sweden; 5: Singlefjord mouth, Sweden; 6: W. Stora Pölsan lighthouse, Sweden; 7: Persgrunden, Sweden; 8: Grisbådarna, Sweden; 9: Kullarna (S.W. Måseskär lighthouse), Sweden; 10: N.W. Hunnebostrand city, Sweden; 11: Gullmarn Fjord, Sweden; 12: Svaberget, Sweden; 13: Väderöarna (Weather Islands), Sweden; 14: W. Ursholmen Island. Names of areas of angling grounds (pos. 7–9 and 12) are based on Lundgren & Waje (2015); currents and depth curves are drawn after Svansson (1975) and Larsson & Stevens (2008).

While the presence of A. palloni along the southern Norwegian coast is clearly documented with sporadic, but repeated, records over the last century, the eastern limits in the Skagerrak are not well established. The species has been noted as not being native to Swedish waters (Pethon & Svedberg, 2004; Nielsen & Svedberg, 2006; Pollard, 2010; Craig & Pollard, 2015), or alternatively, only being present in the Koster Fjord area (Nilsson, 1997; Fig. 1, position 4). Whereas FishBase (http://www.fishbase.org/; Froese & Pauly, 2018) lists the species as native, based on the Swedish checklist of fishes (Kullander, 1999); this checklist, however, lists all species recorded in Swedish waters, including sporadic visitors. In an updated checklist, the occurrence of A. palloni is noted as “sporadic” (Kullander, 2002). The FishBase-associated AquaMaps project (http://www.aquamaps.org/; Kaschner et al., 2016) has a predicted occurrence probability of 0.60–0.79 in Swedish waters (Kesner-Reyes et al., 2012), based on a single verified and a few unverified records in GBIF (the single verified record, from southern Skagerrak, is noted here in the Results section). Recently, the distribution of the species has been suggested to be wider than previously thought in Swedish waters, based on multiple reports of angled specimens.

Materials and Methods

Swedish non-angling records (1993–2016)

Non-angling records were sourced from the scientific literature (Cedhagen & Hansson, 1995; Hallberg, 2011), the Swedish Species Observation System (SSOS; http://www.artportalen.se/), and the GBIF-Sweden Data Portal (http://www.gbif.se/).

Swedish angling records (1995–2016)

The majority of the angling records were obtained from the curated specimen registry (Storfiskregistret) of the Swedish Anglers Association (SAA; http://www.sportfiskarna.se/), where anglers can report catches of fish specimens above a certain species-specific mass-limit, which then gets validated based on photographs, accessory information, and, if needed, expert assessment. The SAA records contains additional information about capture location, depth, habitat, and capture method. The mass-limit for recording an A. palloni in the SAA specimen registry is 250 g (effective since 2012; before that it was 300 g, but no registered records exist from this time-period).

Additional records, were supplied by Swedish anglers, located through posts on internet blogs or through personal communication. A number of records are also direct personal observations by the authors (e.g., specimens in Fig. 2). Furthermore, some records in the SAA registry had limited capture information. In such cases, the angler was contacted.

Figure 2 Pictures of Acantholabrus palloni from Swedish waters (Position 9 in Fig. 1).

(A) Record #8, Table 1; (B) Record #9, Table 1. Published with permission (Photo credit: M. Lundgren).

All records from the SAA database in Table 1 have been verified by the authors from photographs. SAA carries digital copies of all fish in their records. Personal communications were obtained from experienced sea anglers; some of these records are unverified (see Table 1) and should therefore be used mainly as auxiliary information. While misidentification of the species is possible, specimen sea anglers are typically examining their catch closely when resembling a rare species. Hence, the unverified angling records are judged to be valid.

Table 1 Records of A. palloni in Swedish waters.

Record number	Date	Size	Capture method	Location (Fig. 1)	Notes	Information source	
1	1993	Juvenile	Dredge haul	Pos 5	First record from Swedish waters, first inshore record	Cedhagen & Hansson (1995)	
2	July 1995	L: 23 cm M: 142 g	Angling	Pos 8		Hanefors (1995)	
3	09 July 2007	L: No record M: 265 g	Angling	Pos 7	Verified by M. Lundgren	Records of Kungsbacka Angling Club	
4	16 November 2007	L: No record M: No record	Filmed, remotely operated vehicle	Pos 7	50 m depth. Reported by A. Tullot (record #61250199). Unverified	https://artportalen.se/	
5	2008	L: No record M: No record	Unknown	Pos 6	Verified by S.O. Kullander, Swedish Museum of Natural History in GBIF	http://www.gbif.se/	
6	No info. (Pre-2011)	L: No record M: No record	Filmed, remotely operated vehicle	Pos 4		Hallberg (2011)	
7	2010	L: No record M: No record	Angling	Pos 9	Verified by M. Lundgren, direct observation	M. Durell, 2017, personal communication	
8	04 April 2011	L: No record M: 220 g	Angling	Pos 9	Figure 2	M. Lundgren, 2017, personal observation	
9	05 June 2011	L: No record M: 180 g	Angling	Pos 9	Figure 2	M. Lundgren, 2017, personal observation	
10	05 June 2011	L: No record M: 160–180 g (estimated)	Angling	Pos 9		M. Lundgren, 2017, personal observation	
11	26 May 2012	L: No record M: 120 g	Angling	Pos 9		http://www.sg-zander.se/	
12	01 June 2014	L: 26 cm M: 275 g	Angling	Pos 9	12.5 m depth1, rocky bottom	http://www.sportfiskarna.se/, Hellenberg (2014a, 2014b)	
13	26 July 2014	L: 26 cm M: 260 g	Angling	Pos 10	44 m depth, rocky bottom	http://www.sportfiskarna.se/, Hellenberg (2014b)	
14	16 August 2014	L: 26 cm M: 250 g	Angling	Pos 10	47 m depth, rocky bottom	http://www.sportfiskarna.se/, Hellenberg (2014b)	
15	16 August 2014	L: 27.5 cm M: 282 g	Angling	Pos 10	46 m depth, rocky bottom	http://www.sportfiskarna.se/, Hellenberg (2014b)	
16	13 July 2015	L: No record M: 200 g	Angling	Pos 11	First record from the fjord Gullmarn, second inshore record from Swedish waters. Verified by M. Lundgren, direct observation	M. Jonsson, 2017, personal communication	
17	17 July 2015	L: 29 cm M: 296 g	Angling	Pos 12	26 m depth, rocky bottom	http://www.sportfiskarna.se/	
18	06 August 2015	L: 28 cm M: 293 g	Angling	Pos 10	50 m depth, rocky bottom	http://www.sportfiskarna.se/	
19	06 August 2015	L: 28 cm M: 285 g	Angling	Pos 10	50 m depth, rocky bottom	http://www.sportfiskarna.se/	
20	06 August 2015	L: 27 cm M: 267 g	Angling	Pos 10	50 m depth, rocky bottom	http://www.sportfiskarna.se/	
21	09 August 2015	L: 27.5 cm M: 260 g	Angling	Pos 13	35 m depth, rocky bottom	http://www.sportfiskarna.se/	
22	20 August 2015	L: 26.5 cm M: 260 g	Angling	Pos 9	42 m depth, rocky bottom	http://www.sportfiskarna.se/	
23	20 August 2015	L: 26 cm M: 250 g	Angling	Pos 9	41 m depth, rocky bottom	http://www.sportfiskarna.se/	
24	21 August 2015	L: 27 cm M: 260 g	Angling	Pos 13	32 m depth, rocky bottom	http://www.sportfiskarna.se/	
25	22 August 2015	L: 27 cm M: 280 g	Angling	Pos 10	40 m depth	http://www.sportfiskarna.se/	
26	22 August 2015	L: 27.5 cm M: 267 g	Angling	Pos 10	38 m depth, rocky bottom	http://www.sportfiskarna.se/	
27	22 August 2015	L: 28 cm M: 270 g	Angling	Pos 10	40 m depth, rocky bottom	http://www.sportfiskarna.se/	
28	22 August 2015	L: 28.5 cm M: 300 g	Angling	Pos 10	45 m depth, rocky bottom	http://www.sportfiskarna.se/	
29	09 August 2015	L: No record M: No record	Filmed, remotely operated vehicle	Pos 6	30–35 m depth, rocky bottom	Palmkvist et al., 2016	
30	04 October 2015	L: 27 cm M: 270 g	Angling	Pos 6	28 m depth, rocky bottom	http://www.sportfiskarna.se/	
31	24 July 2016	L: 26 cm M: 260 g	Angling	Pos 8	37 m depth, rocky bottom verified from photograph by J. Näslund	A. Enemar, 2017, personal communication	
32–37	September 2016	L: No record M: 70–200 g	Angling	Pos 14	5 individuals. 35–50 m depth, rocky bottom	A. Enemar, 2017, personal communication	
38	19 August 2016	L: 28.5 cm M: 320 g	Angling	Pos 9	42 m depth, rocky bottom	http://www.sportfiskarna.se/ Anonymous (2017a)	
39	25 August 2016	L: 26.5 cm M: 270 g	Angling	Pos 12	41 m depth, rocky bottom	http://www.sportfiskarna.se/ Anonymous (2017b)	
Notes:

1 Possibly an error in the report of depth.

L, total length; M, wet mass.

Results

Swedish non-angling records

The occurrence of A. palloni in Swedish waters was rarely reported prior to 2010, with a first record of a juvenile specimen from year 1993 from somewhere between 50 and 115 m depth in the mouth of the Singlefjord, northeastern Skagerrak (Cedhagen & Hansson, 1995; Fig. 1, position 5). The species has also been previously reported from the Koster Fjord area around the Koster Islands, northeastern Skagerrak (Fig. 1, position 4; Hallberg, 2011), and six km west of Rörö Island (Fig. 1, position 6), southern Skagerrak in 2008 (data provided by Swedish Museum of Natural History, Stockholm; accessed through GBIF-Sweden Data Portal, 25 April 2014; also reported as capture site by Palmkvist et al., 2016). Only two unverified observations were found in SSOS (as of 22 May 2018). The first record comes from Persgrunden (Fig. 1, position 7), stemming from a scientific transect investigation using a remotely operated vehicle in 2007; the second is a questionable record from the harbor of Käringön. The latter record (record 54966014 in SSOS) is not reported in Table 1, due to A. palloni not being a typical species seen in the shallow harbor waters (in contrast to the similar looking goldsinny wrasse Ctenolabrus rupestris (L.)).

Angling records

Between the first Swedish record in 1993 (Cedhagen & Hansson, 1995) and 2011, a few angling records of A. palloni were noted from different sites on the Swedish west coast (Table 1). Between 2011 and 2016, several records of A. palloni have been provided by leisure anglers (Table 1). Repeated captures of the species have been made across years, at least at a few positions (e.g., position 9; Table 1).

Most angled specimens are reported to be caught on, or directly above, rocky bottom at depths of 28–50 m. Another specimen standing out from the rest is record #16 which is the only one caught inshore (in the Gullmarn Fjord, Fig. 1, Position 11), apart from the first Swedish record by Cedhagen & Hansson (1995).

Discussion

The presented records extend the knowledge about the marine ichthyofauna of eastern Skagerrak, which is a generally well documented area regarding fish species distributions (Kullander et al., 2012). In the light of the present summary of these records, we show that angling databases can be utilized as a source for information about the distribution of fish species which are seldom targeted by commercial fisheries, but specifically targeted by anglers.

Acantholabrus palloni belongs to the Swedish ichthyofauna

The angling records of A. palloni show that several individuals of this species being repeatedly caught in the same general locations, at multiple sites, in Swedish waters. It should be noted that number of angled specimens at any given position are likely related to the specific fishing pressure at that site, and data are limited to adult specimens as a consequence of the size-restrictions in the SSA database (see Materials and Methods). Still, these repeated captures across several years indicate that A. palloni could be currently established in, at least some, areas of eastern Skagerrak. Notably, the sites at which the species is recorded, matches the 40–80 m depth curve in Skagerrak, as well as the currents from the North Sea which bring higher-salinity water into the southern Skagerrak and northward along the Swedish coast (Svansson, 1975; Fig. 1). It is worth noting that the current list of records (Table 1) is not a complete record of angled A. palloni, as several other specimens (typically smaller ones) have been verbally described to the authors by anglers, without any specific information being noted by the angler.

Spatial and temporal distribution of the population

The typical capture site for A. palloni is rocky reefs located largely within the general 40–80-m depth range along the Swedish Skagerrak coast. Capture sites are also matching the route of the main currents bringing water from the North Sea. The bottom layer of the deeper parts of Skagerrak have a salinity similar to the North Sea and is substantially more saline than the water originating from the Baltic Sea in the upper layers and in Kattegatt to the south of Skagerrak (Svansson, 1975). This likely makes the conditions in these areas suitable for marine species with a main distribution area in more saline waters, such as A. palloni.

The presence of the species at the offshore islands of the Atlantic (the Azores, Canary Islands, Madeira and Cape Verde) suggests that there is capacity for dispersal in the species. However, the temporal aspect of dispersal appears largely unknown. The angled specimens are all captured between May and October, which represents the time when the angling activity is highest. Given paucity of data from the winter-months, we cannot exclude that the species is a seasonally migratory species in this geographic area. An additional unverified underwater observation was made in November during a scientific expedition at Persgrunden (record #4; Table 1); but November-temperatures at >25 m depth largely matches spring- to early-summer temperatures (Svansson, 1975), so such an observation does not indicate a non-migratory behavior. However, given the apparent strong reef-association, the small body size, and the labriform swimming mode, which is relatively inefficient for long-distance swimming (Sfakiotakis, Davies & Lane, 1999), a seasonal migratory life-style seems unlikely. Notably, Craig & Pollard (2015) states that it is not a migratory species; however, without explicit support from data or references.

Stability of the population

The first Swedish record was a juvenile individual (Cedhagen & Hansson, 1995), which is indicative, but not proof, of spawning in the area. It could be possible that Skagerrak acts as a sink for the A. palloni population, with fry or young individuals drifting into the Skagerrak area from the North Sea and then settling on suitable rocky reefs. Until spawning and viable fry can be demonstrated from Skagerrak, it is not possible to know whether the Skagerrak population is self-reproducing. Systematic investigations of angled specimens (e.g., in association with marine angling competitions) could provide information about spawning activity, as well as other aspects of their biology, such as feeding habits and parasite fauna—all of which are largely unknown for the species (Kullander et al., 2012). Studies on population genetics may provide further insights into the population structure of the species (Faust et al., 2018). Hence, such investigations could be encouraged to extend the knowledge of this data-deficient species.

Range extension due to climate change?

The recent surge in records could be an indication of a range extension. One hypothetical factor that could lead to range extensions is the changing climate which leads to higher seawater temperatures (Dye et al., 2013). While increased sea water temperature was also noted from 1939 to 1960, the warming effect has been particularly strong in the north–east Atlantic during the last decades (Dye et al., 2013). The estimated temperature increase in Skagerrak, from 1990 to 2014, is 1.6 °C (Rinde, Hjermann & Staalstrøm, 2016). Range extensions has indeed been indicated in several European marine areas, including Scandinavian waters (Hiddink & Ter Hofstede, 2008; Nicolas et al., 2011; Brattegard, 2011). Brattegard (2011) estimated that more than 500 marine benthic species expanded their range northwards in Norwegian waters, in the range of 750–1,000 km, between 1997 and 2010. Furthermore, several new species were found to establish along the Norwegian Skagerrak coast during this time (Brattegard, 2011; Rinde, Hjermann & Staalstrøm, 2016). Even though the angling records are located south of the previous northern range limit, the eastern Skagerrak is still at the edge of the species’ northern distribution range (AquaMaps presents an estimated native distribution map that extends to the Faroe Islands, but this is not supported by actual records of the species; Kaschner et al., 2016; Kesner-Reyes et al., 2012). Hence, a climate-related population increase at the northern distribution range may cause spillover effects into the Swedish waters of Skagerrak, following the main currents in this area (cf. Fig. 1 and Rinde, Hjermann & Staalstrøm, 2016). However, the paucity of historical data on the species makes it impossible to conclude whether this is the case or not. The fact that the species was recorded in south-western Norway already in 1869 (Öberg, 1870), with further observations along the south coast of Norway noted prior to 1942 (Andersson, 1942), suggests that it was present, at least temporarily, close to Swedish waters prior to the recent increase in the seawater temperature. It is also very hard to distinguish climate effects from other effects such as fishing or meta-population dynamics (Brander & Havenhand, 2016).

Prior records missing due to historical angling methods and oversight?

Acantholabrus palloni is generally captured over off-shore rocky reefs. Smaller-sized marine fish species, such as A. palloni, have historically not been targeted by off-shore anglers to the same extent they currently are (Lundgren & Waje, 2015). While coastal anglers use a wide variety of hook-sizes, off-shore anglers have, until recently, mainly used larger hooks targeting larger species. With the relatively small gape-size of A. palloni, this may effectively have eliminated the species from being caught, even though the capture sites reported here have been commonly targeted reefs for angling (Lundgren & Waje, 2015). In addition, the typical rocky reef habitat is likely seldom trawled by commercial fishermen, probably leading to few specimens being caught in fisheries overall (Pollard, 2010). While clearly distinguished at closer inspection (which is typically what specimen sea anglers do), it also resembles the common goldsinny wrasse and young female cuckoo wrasse Labrus mixtus L. in coloration, which may lead to misidentification and oversight by, for example, fishermen not interested in the species for commercial purposes. These facts may have limited the number of captures and historical reports from Swedish waters. Similar oversight might apply to other rare Scandinavian species, such as Thorogobius ephippiatus (Lowe, 1839) and Pomatoschistus norvegicus (Collett, 1903) (Holm & Mattson, 1980; Cedhagen & Hansson, 1995).

Value for Swedish fisheries

Despite its relatively small size, A. palloni has value for marine specimen angling, for example, in marine angling competitions where the number of species caught is rewarded. In fact, the angling records being presented in this article are largely a consequence of this species being acknowledged as a target for specimen anglers, who are specifically targeting large specimens of different species (Hellenberg, 2014a, 2014b; Lundgren & Waje, 2015). In commercial fisheries, however, it has generally little or no value (Machias et al., 2001; Nielsen & Svedberg, 2006), although it has been noted among the targeted species in some Mediterranean countries (Alegre, Lleonart & Veny, 1992; Economidis & Koutrakis, 2001). Smaller wrasse species are fished commercially in Scandinavia for sea lice control in salmon farms (Espeland et al., 2010), but A. palloni is not suited for such fishery as it is deeper-living than most of the other smaller wrasses (e.g., corkwing wrasse Symphodus melops (L.) and juvenile ballan wrasse Labrus bergylta Ascanius, 1767) and, thus, generally subjected to severe barotrauma when hauled, resulting in inflated swim-bladders and bulging eye-balls (see, e.g., Fig. 2).

Using angling records for conservation and management of rare fishes

This study presents a case where citizen generated data can be used for increasing the knowledge about a rare species. It is possible that recognition of the angler community as a valuable source of information may increase anglers’ interest in continuing and increasing the frequency and detail of their reports. Anglers are often keen to participate in the management of fish stocks, but may also be skeptical about revealing the exact location of their fishing sites, so this type of recognition may lead to a further increased interest in fish conservation issues (Granek et al., 2008; Fetterplace et al., 2018). From a conservation and management perspective, the current data provides a tentative picture of the distribution and habitat in Swedish waters. A quantitative habitat suitability model (HSM) is, however, not possible to construct, mainly due to the scarcity of data about habitat conditions at capture sites. Bradter et al. (in press) constructed HSMs for Siberian Jays in Sweden, utilizing data from the SSOS. As compared to, for example, birds, for which the SSOS reporting system is run in collaboration with BirdLife Sweden (the Swedish Ornithological Society), fish appear to be under-reported in SSOS. Hence, the SAA database appears to be substantially more informative and reliable for observational reports of fish at the present point in time, despite having a lower mass-limit for reports, which limits the number of reports in general and completely eliminates reports of juvenile specimens.

Conclusions

In this article, we summarize the present knowledge about the current distribution of A. palloni in Swedish waters, at the edge of the distribution range of the species. The records of A. palloni presented here in particular highlight the importance of citizen-generated data in the form of anglers’ reports and angling records as useful contributions for ichthyological investigations of presence and distribution of non-commercial fish species.

We thank Magnus Durell, Mattias Jonsson, Mattias Liewendahl, Arvid Enemar, and Dan Calderon for providing details on their catches of scale-rayed wrasse. Nicka Hellenberg is thanked for curating the Storfiskregistret specimen database. Three anonymous reviewers are thanked for contributing valuable comments and information.

Additional Information and Declarations

Competing Interests

Author Contributions

Field Study Permissions

Data Availability

The authors declare that they have no competing interests.

Joacim Näslund conceived and designed the experiments, analyzed the data, prepared figures and/or tables, authored or reviewed drafts of the paper, approved the final draft.

Markus Lundgren conceived and designed the experiments, contributed reagents/materials/analysis tools, authored or reviewed drafts of the paper, approved the final draft.

The following information was supplied relating to field study approvals (i.e., approving body and any reference numbers):

Fish specimens reported were caught following Swedish or Norwegian angling regulations. All cases where the authors captured recorded specimens were part of non-scientific angling expeditions, conducted prior to the conception of the study.

The following information was supplied regarding data availability:

The raw data is contained in Table 1.

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
