# Peer review of "Mapping the distribution of scale-rayed wrasse Acantholabrus palloni in Swedish Skagerrak using angling records"

_PeerJ, doi:10.7717/peerj.5900_

## Round 0.1 · original submission · Major Revisions

The reviewers have identified a number of relevant issues. I tend to agree more with reviewer 2 and suggest a major revision given that the conclusions are supported by the data. However, the introduction demands a proper repositioning and re-write; it does not follow scientific standards and lacks relevant details. In the Results I would like you to plot also the native areas and ideally try to track the "invasion" to your study site better. I think ichthyologists are increasingly using the approach you support and in fact I know from key scholars they are nowadays tracking facebook to identify observations of non-native fishes. In that sense your manuscript is valuable as a poof of concept, but the ecology and biogeography of the species should be better integrated with your study area.

Reviewer 1 ·

Basic reporting

I have reviewed the manuscript “Mapping the distribution of scale-rayed wrasse Acantholabrus palloni in Swedish Skagerrak using angling records “.
The manuscript maps the spatiotemporal occurrence of scale-rayed wrasse in the Skagerrak area. The authors use underutilized angling records for this matter, which could act as method in coastal monitoring, e.g. for inclusion in MSFD monitoring programs, mapping presence/absence of endangered species. However, I question the utility of the study as there is no clear research objective, structure and discussion of the manuscript.

Experimental design

In general, the manuscript provides little new knowledge on the occurrence of scale-rayed wrasse and/or the use of angling records as a monitoring methodology. Furthermore, the manuscript does not fall within the scope and aims of PeerJ concerning the objective determination of scientific and methodological soundness (see comment above). I would recommend to reject the submitted manuscript.

Validity of the findings

The authors state that the species has been noted as not being native to Swedish waters. However, fishbase.org attests a probability of occurrence of this species in the Skagerrak of 0.6-0.79 and lists the scale-rayed wrasse as native species in SE since 1999. The same source also attests a northern distribution range further up the Norwegian coast unlike the authors conclusion that the occurrence in the Skagerrak areas maps the edge of the species northern distribution.
In the case of historical records, the data sources used could not be verified, e.g. GBIFs data provides no picture. The authors recognize that the mass limit for recording (250-300g) and the small gap size of this species compared to the regular hook and bait sizes used skew the list of records towards larger specimen. Unfortunately, the manuscript fails to address this major limitation in the discussion as these factors not only skew the distribution to larger individuals but may also lead to complete non-reporting of the occurrence of this species as standard angling gear used in these depths is not suitable to capture this relatively small species (25cm ~ 183g according to size and LWR data from fishbase).
Why was no comparison done with commercial sampling data (coastal monitoring programs)? Without comparison it is difficult to be certain about the utility of using angling records for monitoring purposes.

·

Basic reporting

Please see my comments to authors, but in summary

English is ok
More references especially on angler citizen science could be usefull
Article structure is not super, but can easily be improved.

Experimental design

Please see my comments to authors

There are some issues with the methods and replicability,i.e. in terms of using personal communications as a significant part the data material. But I think the authors can improve this

Validity of the findings

Please see my comments to the authors
IN short I find the conclusion supported by the data

Additional comments

This study use a citizen science approach, i.e. catch records form anglers, to demonstrate that A. palloni is present in areas near the Swedish coast where it has not previously been found. The study therefore give a noce example of how angler citizen science can be a useful contribution for ichthyological investigations, i.e. spatial patterns, of non-commercial fish species.

I find the study relevant and innovative, but I have also some concerns as well as suggestions for improvements that I hope the authors would consider.

In my opinion, the study is based on angler citizen science. Hence in the introduction, I miss a brief general introduction to angler citizen science as such, e.g. how common it is, some examples and its potential, e.g. to describe presence of rare species. I would put this at the very beginning of the introduction as this would lead to the specific aim of the study, i.e. to demonstrate presence of rare species.

Further the introduction lacks study objectives, i.e. “this study aim to demonstrate how angler driven citizen science e.g. through web-based log books where anglers report catches and internet scraping, can further our knowledge about presence of rare species”….or something along those lines.

IN terms of replicability, I see it as a fundamental problem, when “personal communications” and “own observations” constitute a major part of the results. Depending on how you view it, such input can be hard to replicate. In this case it was 12 out of 37 observations i.e. close to 33% of the observations. Still I judge that the personal communication results seems consistent with the rest of the data, so I tend to say it is ok. But this potential problem should be acknowledged in the discussion and supplemented by statements about how you verified these “personal communication” reports.

You discuss temporality over years, i.e. if your results suggest that A. Palloni arrived recently to the areas, or if it has been present the whole time, but anglers have not fished for them. I think that you can not conclude anything about temporal patterns, so a conservative approach would be to conclude that the study simply demonstrate that A palloni is now present in the area (and probably well established), but the study can not be used to illucidate any temporal patterns.

On the other hand you do not dicuss temporality within a year, which I suggest you consider to do. In table 1 clearly more reports are from the summer period, which probably reflects that this is the time of year where the anglers fish for A. palloni. But maybe there is reports of anglers fishing outside summer without luck that could allow discussions/speculations about whether A. palloni migrate in and out of the area on a seasonal basis? Just a suggestion.

At several places in the introduction, the methods and results the authors tend to engage in discussions. This should be moved to the discussion section. See specific examples below.

Having said all this I think that the wording of the conclusion presented in the ms is very precise and could be used as a guideline for the authors during their revision. The wording in the conclusion contains pretty exact what can be concluded from the study, i.e. that angler citizen science in terms of angler reports and records (whats the difference btw?) can be a useful contribution for ichthyological investigations of non-commercial fish species. I agree with the conclusion and think iits supported by the data presented. Therefore, if the paper is properly revised, it can stand as a nice illustration of the usefulness of angler citizen science.

To sum up
The Introduction needs more work and should include study objectives. A lot of the present text in intro, rmethods and results should be moved to the discussion. Finally the discussion should be expanded (to include a discussions of;

-Temporal stability (between years and within years)

-Reliability of the data and shortcommings, including the role of personal communication data in the overall study and how they were verified, the likelihood that even more fish were present but not sampled/presented (which is now placed in methods and results e.g. lines 64-67 & 98-100), did anglers misidentify and there is probably more.

-Some of the present discussion can be integrated and reused, but I would suggest a rearrangement of the discussion into a more clear structure.
The conclusion is nice and clear 

Specific comments
Line 41-45. Move this to the discussion, where you can use it to argue that chances of misidentification was low. Having said that you argue on line 118-120 that misidentification is indeed possible. So please be clear about this in the discussion.
Lne 64-67. This should be moved to the discussion section, where it can be included in a section discussion shortcomings of the study, e.g. that anglers reporting is skewed towards larger specimen as they probably do no capture the smaller ones due to gear choice.
Line 72-73 you state that all records was verified by the authors. Please explain how you did this in case were data was provided as “personal communications”?
Line 90-92. This should be moved to the discussion as a part of where you discuss temporality, i.e. if your results suggest that A. Palloni arrived recently to the areas, or if it has been there the whole time but anglers have not fished for them. I think that you can not conclude about this, so a conservative approach would be to conclude that the study simply demonstrate that A palloni is now present in the area, but the study can not be used to illucidate any temporal patterns. IN other words it is likely that the sudden emergence of A.palloni in the angler reports is a result of behavioral changes of anglers towards being specimen fishers and that the word of A. palloni as a potential species have been spread. Hence angling pressure have increased and more A. palloni have been reported
Line 98-100. This should also be moved to the discussion and used to support that the presented results are likely to be conservative, i.e. that you have received personal communication reports about even more A. Palloni, but not included these as they did not contain sufficient information.
Line 118-120. In the introduction you state that misidentification is unlikely…but here you state the opposite. Please expand this discussion and be precise. Is misidentification likely or not and how could if have affected your data.

---

## Round 0.2 · Major Revisions

I think you have properly addressed the reviewers. Only smaller issues remain as identified by the reviewer.

Reviewer 1 ·

Basic reporting

The authors have done a very good job to improve the quality of the manuscript, e.g. adding a clear study objective, adding context to the issues related to data collection of data-limited species, specifying the genesis and records of the first occurrence of A. palloni in Swedish waters, widening the discussion to include seasonal occurrence, issues on species identification, and the use of citizen science data.

The issue on species occurrence and the different standards used by the different data sources (e.g. fishbase vs SAA) were addressed by the authors in the response to the reviewers and partly in the revised manuscript. In the response to the reviewers the authors give a more thorough description on the limitation of fishbase as distribution mapping database. This should be included in the discussion as this is useful information to judge the validity and quality of such data. I leave this decision however with the editor.

In the text I have identified a few typos, and made a comment where the wording is vague or sentences are too long. After these small changes I suggest the article is ready to be published.

Experimental design

no comment see above

Validity of the findings

no comment see above

Annotated reviews are not available for download in order to protect the identity of reviewers who chose to remain anonymous.

Reviewer 2 ·

Basic reporting

Experimental design

Validity of the findings

Additional comments

Review of “Mapping the distribution of scale-rayed wrasse Acantholabrus palloni in Swedish Skagerrak using angling records”.

I got this manuscript after revision by other reviewers. I have read the comments given by them and agree with much of the critizism.
The manuscript gives an overview of the knowledge of a fish species that is poorly known and difficult to study. It shows that it is not an occasional guest but a species that really belongs to the Swedish fish fauna. The manuscript is relevant and interesting and I recommend it to be published with revisions. However, I had problems with the DISCUSSION. It must be structured in a better way. I have made an analysis of the current structure (contents of each paragraph) and give suggestions for improvements. Remember, one paragraph should in principle contain only one aspect or topic. Some of the paragraphs contain more than one topic and these topics come in a somewhat illogical order.

ABSTRACT
Line 15: Delete: “In this paper, “ and “tentatively”.

DISCUSSION, contents of paragraphs:
§1, line 152: General conclusion/introduction to the Discussion.
§2, line 155: This paragraph is a mixture of different aspects: lines 155-162 and 171-173 contain comment on the results in this study. Lines 162 (Speculatively …) to 171 (… 2016) is a speculation about climate effects but is not based on any references. These aspects should be sorted into different paragraphs.
§3, line 174: Also this paragraph is a mixture of various aspects. Lines 174-176 covers seasonality related to bias within the method; Lines 176… is about an additional record in November and how it is related to the hydrography; Lines 179… is a general discussion about the species’ abilities for dispersal; Lines 189… is about the value and limitations of angling data, as well as potential future studies.
§4, line 197: habitat and recent changes in fishing methodology.
§5, line 211: the value of Acantholabrus palloni for anglers, commercial fisheries and aquaculture.
§6, line 223: the value of citizen generated data and comparison with ornithology.
§7, line 239: existence of additional observations.
My suggestion is that the contents of the discussion is reorganized in the following order:

General conclusion/introduction to the Discussion.
Relate to the results in this study.
Bias in the methodology
Changes in fishing methodology during the last 150 years.
Distribution in the Skagerak area related to the hydrography.
Biogeography of the species (It has an enormous distribution, from Gabon to Norway, and exists also around mid-Atlantic islands (Acores, but also Madeira and Canary Islands, Source: FishBase). This indicates ability for dispersal. The first sample in Sweden (Cedhagen & Hansson 1995) was a juvenile. It must have been born in the vicinity of the sampling station.
“Climate change” is a kind of mantra expressed in various situations, typically without any deep thoughts about it. The warmest decade of the last century, on a global scale as well as in Skagerak, was the 1930’s, but this is seldom reported. (It was ice-free up to the North Pole). A summary of the old oceanographic data in the Gullmar fjord where was published by Svansson (1984). The climate anomaly data during the 1930’s for the Gullmar fjord is available there. • The species was sampled near Flekkefjord, southernmost Norway, already in 1869 (Öberg 1870). This find was referred by Liljeborg (1891) and Stuxberg (1894). It was during the last part of the so called “Little Ice Age”. Andersson (1964, 3. Ed.) wrote that the species was found a few times from Kristiansand to Stavanger (The species was already mentioned in the first edition of this book from 1942). A recent invasion of the species in the region because of climate change during recent time (say the last 40-50 years) can therefore be ruled out.
The authors refer to Kullander & al. (2012). This is a very good book but it does not include all knowledge, or all details, up to that year. The oldest reference in the manuscript is Svanssons oceanography paper from 1975. This is fine, but the oldest reference about fish was published in 1990 and most of the sources are probably accessible on Internet. This is a very limited source of information, particularly when writing about faunal changes in a historical perspective. Young scientists must understand that it is important to consult also older, analogous references in printed form, such as books. The university library in Stockholm, where the first author works, has one of the absolutely most complete scientific libraries on this planet, particularly natural sciences within the period 1880-1960. It can match corresponding libraries in, e.g., London, Paris and St. Petersburg. The authors have no excuse not to use it.
I guess that the lack of previous reports of Acantholabrus palloni could be a parallel to that of the overlooked fish species Thorogobius ephippiatus (Holm & Mattson 1980) from the same area.
References
Andersson, K.A. (Ed.). 1964. Fiskar och fiske i Norden, vol 1. 3. Ed. Natur och Kultur, Stockholm.
Holm, T. & Mattson, S. 1980. Thorogobius ephippiatus (Pisces) found on the west coast of Sweden. Sarsia 66: 87-88.
Liljeborg, W. 1891. Sveriges och Norges Fauna. Fiskarne. Upsala, W. Schultz.
Stuxberg, A. 1894. Sveriges och Norges Fiskar jämte inledning till fiskarnes naturalhistoria, (3): 354. Albert Bonniers Förlag, Stockholm.
Svansson, A. 1984. Hydrography of the Gullmar Fjord : Gullmarsfjordens hydrografi. Meddelande från Havsfiskelaboratoriet, Lysekil, (And references to primary data from the Gullmar fjord since the year 1900 in its references).
Öberg, P.E.W. 1870. Acantholabrus Couchi Cuv. Et Val., en för Skandinaviens Fauna ny fiskart. Öfversigt af Kongl. Vetenskaps-Akademiens Förhandlingar 1870, No 4: 391-395.

---

## Round 0.3 · accepted · Accept

Thanks for taking all the effort to address the reviewers and sorry for the delay. It was hard to get reviewers on this manuscript.

# "Long thought to be occasional vagrants outside its known range in the eastern Atlantic Ocean and Mediterranean Sea, we ask if this rare and understudied labrid has expanded its range and become established in Swedish waters."

This addition will make it clear that your article meets our requirements for asking a biological question, rather than simply reporting on a range expansion. It can be inserted while in production.#